# Short, Lipidated Dendrimeric γ-AApeptides as New Antimicrobial Peptidomimetics

**DOI:** 10.3390/ijms24076407

**Published:** 2023-03-29

**Authors:** Yafeng Wang, Menglin Xue, Ruixuan Gao, Soumyadeep Chakraborty, Shaohui Wang, Xue Zhao, Meng Gu, Chuanhai Cao, Xingmin Sun, Jianfeng Cai

**Affiliations:** 1Department of Chemistry, University of South Florida, 4202 E. Fowler Ave., Tampa, FL 33620, USA; 2Department of Molecular Medicine, Morsani College of Medicine, University of South Florida, 12901 Bruce B Downs Blvd., Tampa, FL 33612, USA; 3College of Pharmacy, University of South Florida, 12901 Bruce B Downs Blvd., Tampa, FL 33612, USA

**Keywords:** dendrimeric γ-AApeptides, broad spectrum activity, antibiotic resistance

## Abstract

Antibiotic resistance is one of the most significant issues encountered in global health. There is an urgent demand for the development of a new generation of antibiotic agents combating the emergence of drug resistance. In this article, we reported the design of lipidated dendrimeric γ-AApeptides as a new class of antimicrobial agents. These AApeptides showed excellent potency and broad-spectrum activity against both Gram-positive bacteria and Gram-negative bacteria, including methicillin-resistant Staphylococcus aureus (MRSA). The mechanistic studies revealed that the dendrimeric AApeptides could kill bacteria rapidly through the permeabilization of bacterial membranes, analogous to host-defense peptides (HDPs). These dendrimers also did not induce antibiotic resistance readily. The easy access to the synthesis, together with their potent and broad-spectrum activity, make these lipidated dendrimeric γ-AApeptides a new generation of antibacterial agents.

## 1. Introduction

It is estimated that 700,000 to several million deaths result from bacterial infections per year, and 2.8 million people are infected by bacteria resistant to current antibiotics in the USA, with at least 35,000 people dying from that. Due to the scarcity of new antibiotics to combat antibacterial resistance (AMR), over 50 million people could die by 2050, with the yearly death toll being 10 million under the current predicted model [1]. As a result, it is extremely urgent to develop new antibacterial agents that can mitigate emerging antibiotic resistance.

Host defense peptides (HDP), which are produced by organisms as the first-line agents to defend against a wide range of bacteria, have gained much attention from scientists [2]. Usually, HDPs share two common features: one is the cationic charges, and the other one is a proper ratio of hydrophobic residues, which enable HDPs to adopt amphipathic structures and exhibit significant selectivity toward bacteria over mammalian cells. This is because the outer leaflet of the mammalian cell membrane is composed of zwitterionic lipids, while the outer leaflet of the bacteria membrane mainly consists of negatively charged phospholipids. As a result, the negatively charged bacterial membrane tends to interact with cationic HDPs preferentially, leading to bacterial cell death while maintaining low hemolysis risks and cytotoxicity [3]. Nevertheless, there are still drawbacks associated with HDPs, such as proteolytic degradation susceptibility, poor selectivity, and moderate activity. To address these problems, several classes of peptidomimetics have been developed to overcome the drawbacks of HDP, including peptoids [4], β-peptides [5], γ-AApeptides [6], and oligourea [7]. These new unnatural peptidomimetic sequences can mimic HDP function against pathogens and retain resistance to proteolytic hydrolysis [8].

Lipidated peptides have been presented as antibiotics for years. For instance, polymyxin B [9] and daptomycin [10] are two FDA-approved lipo-cyclic peptides. While daptomycin only displays activity against Gram-positive bacteria, polymyxin B is only active for Gram-negative bacteria. Despite distinct antibacterial mechanisms, it has been shown that lipid tails are critical for their activity, which facilitate bacterial membrane interaction [11]. Recently, our group designed a new class of antimicrobial peptidomimetic compounds composed of γ-AA amino acid (Figure 1) [6]. To further explore the antimicrobial potential of γ-AApeptides, we herein report a new class of short, lipidated dendrimeric γ-AApeptides as potential antibacterial agents. 

## 2. Results and Discussion

In our current design, a positively charged γ-AApeptide building block was attached to the secondary amine of another γ-AApeptide building block, to which different lengths of lipid tails were introduced to make a series of miniature lipo-dendrimeric γ-AApeptides (Figure 1). As such, these amphipathic structures were expected to mimic the antibacterial function of HDPs. The positively charged side chains would form electrostatic attraction with bacterial membranes, whereas the lipid tail would facilitate the insertion of the compounds into bacterial membranes, leading to membrane disruption. To this end, the compounds were tested for the ability to kill a panel of Gram-positive and Gram-negative bacteria (Table 1). For the building blocks conjugating to the C16 lipid tail, such as Leu (**YW-1**), Phe (**YW-2**), Ala (**YW-3**), Tyr (**YW-4**), Ser (**YW-5**), Lys (**YW-6**), and Arg (**YW-7**), different γ-AA amino acids were synthesized and solid phase was used to do final synthesis, the structure of newly synthesized compounds listed in Figure 1. It shows these compounds generally exhibited effective antimicrobial activity and did not show hemolytic activity up to 125 µg/mL. While **YW-1** displayed the most broad-spectrum antimicrobial activity against a panel of Gram-positive and Gram-negative bacteria, certain compounds, such as **YW-3**, **4**, **6**, and **7**, demonstrated highly selective activity toward MRSA (MIC: 0.75–1.5 µg/mL). The findings suggested that functional groups on the building blocks conjugated to the C16 lipid tail could play an important role in antibacterial activity and selectivity, which could direct the future design and optimization of this class of compounds. 

## 3. Material and Methods

### 3.1. Membrane Depolarization Study

To probe the mechanism of antibacterial activity, we first carried out a membrane depolarization study with the membrane potential-sensitive dye 3,3′-dipropylthiadicarbocyanine iodide (diSC_3_5) (Figure 2A,B) [12]. DiSC_3_5 usually accumulates in living bacterial membranes, and due to self-quenching, it shows low fluorescence intensity. However, if the membrane is disrupted, and the fluorophore is released from the membrane, fluorescence will improve dramatically. As shown in Figure 2, when the bacteria were treated with different concentrations of the lead compound **YW-1**, both *E. coli* and *MRSA* exhibited dose-dependent increases in fluorescence intensity. It was intriguing that **YW-1,** at a two-fold of MIC or above, caused even more intensive fluorescence than the positive control Triton, which is well known for causing membrane damage. This experiment result demonstrated that the bacterial membranes were disrupted by **YW-1**, indicating a potential mechanism similar to that of HDPs.

### 3.2. Outer Membrane (OM) Permeabilization

Next, using *E. coli* as a microorganism, we evaluated the OM permeabilization of **YW-1**. In aqueous conditions, 1-N-Phenylnaphthylamine (NPN) is blocked by the cell wall. However, if the OM is permeabilized, and NPN is taken up as a result, the fluorescence intensity will increase, compared to non-treated OM [13]. As shown in Figure 3A, with 1% Triton as a control, the permeabilization capability of the OM was determined by the absorbance of NPN in a concentration-dependent manner. **YW-1** exhibited a good potency in outer membrane permeability with 1 × MIC of 98%, compared with that of 1% Triton.

### 3.3. Inner Membrane (IM) Permeabilization

We next used the o-nitrophenyl-β-d-galactopyranoside (ONPG) hydrolysis assay to test the ability of the lead compound **YW-1** to permeabilize the inner membrane of Gram-negative bacteria (Figure 3B). If the lead compound **YW-1** compromised the bacterial inner membrane, ONPG would interact with the cytoplasmic enzyme β-galactosidase to form o-nitrophenol, which can be measured under OD 420 nm [14]. As shown in Figure 3B, when *E. coli* were treated with **YW-1** with 8 × MIC, 4 × MIC, 2 × MIC, and 1 × MIC, the OD 420 nm intensity was enhanced to a great extent. 

### 3.4. Fluorescence Microscopy

To further assess that the bacteria were killed with **YW-1** by comprising the cell membrane, we conducted a fluorescence microscopy experiment. Two dyes, 4′,6-diamidino-2-phenylindole (DAPI) and propidium iodide (PI), were employed in this experiment. PI can only stain dead or injured cells, since it does not have cell membrane permeability. Instead, DAPI can be used to stain both dead and live cells because it is cell-permeable. After MRSA and *E. coli* were treated with **YW-1**, red fluorescence was observed under the PI channel (Figure 4), suggesting that bacterial membranes were compromised by **YW-1**. 

### 3.5. Transmission Electron Microscopy (TEM)

TEM microscopy offers convenient access to check the intactness of cell membranes. The cell membranes of MRSA and *E. coli* were intact. However, after treatment with **YW-1**, the membranes clearly demonstrated damages, and bacterial cells lost their spherical (MRSA) or rod (*E. coli*) shapes (Figure 5), indicating that the membranes of these bacteria were disrupted. 

### 3.6. Bacterial Killing Efficiency

#### 3.6.1. Time-Kill Kinetics Study

To evaluate how rapidly bacteria could be eradicated, the lead compound **YW-1** was examined for its ability to eradicate MRSA and *E. coli* by studying the time-kill kinetics. As shown in Figure 6, **YW-1** could remove MRSA and *E. coli* thoroughly with 4 × MIC and 8 × MIC within 2 h. 

#### 3.6.2. Drug Resistance Test

Since the abuse of antibiotics, antibacterial drug resistance has become an increasingly serious threat. Thus, it is of great importance to assess the drug resistance of newly developed lead compounds. As shown in Figure 7, the control, ciprofloxacin, developed drug resistance after 14 generations, since the MIC increased about 100-fold. In comparison, **YW-1** did not have an obvious change in MIC, which suggests that **YW-1** has a low probability of developing antibiotic resistance. Our compound had a lower probability of developing drug resistance.

### 3.7. Experiment Procedure

#### 3.7.1. General Experiment Methods

Chemical reagents and solvents were bought from Sigma Aldrich, Oakwood, TCI, and Chem-Impex. Final products were purified with Waters Breeze 2 HPLC and lyophilized using Labconco lyophilizer. HPLC traces of the final product were collected using 5–95% acetonitrile in water with 0.1% HPLC-grade TFA for at least 40 min. The nuclear magnetic resonance (NMR) data were collected using the Agilent 600 MHz NEO instrument. High-resolution mass spectra of compounds (Appendix A) were collected using an Agilent Technologies 6540 UHD accurate-mass Q-TOF LC/MS spectrometer. Antibacterial assays and mechanism of action studies were performed using a Biotec multimode microplate reader synergy H4. Six different species of bacteria were used for bacteria assay tests, such as MRSA (ATCC 33591), MRSE (RP62A), VREF (ATCC 700802), *E. coli* (ATCC 25922), *K. pneumoniae* (ATCC 13383), and *P. aeruginosa* (ATCC 27853).

#### 3.7.2. MIC (Minimum Inhibitory Concentration)

Concisely, *E. coli*, *K. pneumoniae*, *P. aeruginosa*, MRSA, MRSE, and VREF were cultivated in 37 °C TSB medium for 16 h. Subsequently, 4 ml of new TSB medium was added to 100 μL of cultivated bacterial solution and incubated for another 6 h to reach the mid-log phase. Next, 96-well plates were injected with 50 μL of bacterial solution that reached mid-log phase and 50 μL of compounds with different concentrations, which ranged from 0.75–25 μg/mL. Then, the 96-wells plates were incubated for another 16 h at 37 °C. After 16 h, MICs were determined by a multimode microplate reader.

#### 3.7.3. Hemolytic Activity

Human red blood cells were washed with 1 × PBS buffer, then centrifuged at 700 g for 10 min. After discarding the top clear solution, the bottom cells were diluted to obtain 5% solution using 1 × PBS buffer.

Then, 50 μL of cell solution was injected into 96-well plates. Following this, 50 μL of synthesized compounds with different concentrations from 250 to 1.95 μg/mL were added into 96-well plates. Then, the mixture was incubated for 1 h at 37 °C. After centrifuging at 3500× *g* rpm for 10 min, 30 μL of the supernatant was transferred into a new 96-well plate with 100 μL of 1 × PBS buffer. With the same microplate reader, the data of absorbance at 540 nm were compared. Positive control: 2% Triton-100. Calculation formula: percentage of hemolysis = [(Absorbance of sample-Absorbance of PBS)/(Absorbance of Triton-Absorbance of PBS)] × 100

#### 3.7.4. Drug Resistance Study

The first generation of MIC data of **YW-1** against MRSA and *E. coli* was already obtained in an MIC study. Then, the MRSA of the first generation in the well next to the last clear well was diluted to the mid-log phase, and MICs were tested at 37 °C. This step was repeated for 14 passages. The data of drug resistance for *E. coli* were obtained using the same method with bacterial *E. coli*.

#### 3.7.5. TEM

An amount of 30 μL of mid-log phase MRSA and *E. coli* was diluted to 3 mL in TSB medium with 2 × MIC of **YW-1**, and the mixture was incubated for 2 h. The bacterial pellets were centrifuged at 3000× *g* rpm for 10 min. For the next step, PBS buffer was used to wash three times. Then, the suspended bacterial samples were dropped on grids and dried in a vacuum oven at 45 °C. TEM images were obtained using a FEI Morgagni 268D TEM, with an Olympus MegaView III CAMERA on the microscope, at 60 kV.

#### 3.7.6. Inner Membrane Permeability

Mid-log phase *E. coli* was obtained in Mueller Hinton Broth with 2% lactose at 37 °C; then, it was centrifuged at 3000× *g* rpm for 10 min at 4 °C and washed with 20 mM glucose and 1.5 mM ONPG in 5 nM HEPES buffer one time. Next, the bacterial solution was diluted until OD_600_ = 0.1 using the same buffer. Following this, 50 μL of diluted bacterial solution was injected into a 96-well plate, and 50 μL of **YW-1** in different concentrations and melittin were injected into the bacterial solution, respectively. The OD_420_ was read at 37 °C every 6 min until the fluorescence reached the highest plateau.

#### 3.7.7. Fluorescence Microscopy

A total of 30 μL of mid-log phase bacterial solution was diluted to 3 mL in TSB medium with 2 × MIC of **YW-1**. The mixture was incubated for 2 h at 37 °C. Following this, the bacterial solution was centrifuged at 3000 rpm for 10 min. The top solution was thrown away, the bottom bacterial pellets were washed with PBS buffer, and PI (5 μg/mL) and DAPI (10 μg/mL) were added sequentially on ice under a dark environment. After dyeing with PI and DAPI, the bacterial cells were washed with PBS buffer. Immediately after 100 μL of PBS was added to suspend the bacterial cell, 10 μL of the suspended solution was dropped onto the slide, and data were obtained using a Zeiss Axiovert 200 inverted microscope.

#### 3.7.8. Time-Kill Kinetics Study

Different concentrations of **YW-1** and ciprofloxacin were mixed with 300 μL of mid-log phase bacterial solution in TSB medium. The mixture was incubated for 0, 10, and 30 min and 1 and 2 h, respectively. At the time, *E. coli* were diluted 100-fold, and the MRSA was diluted 100-fold. A total of 100 μL of each was transferred on TSB agar plates. After 16 h at 37 °C, CFUs were read using the Biotec multimode microplate reader.

## 4. Conclusions

In summary, we developed a new series of antibacterial compounds. The lead compound, **YW-1**, showed potency towards both Gram-positive and Gram-negative bacteria. Further, **YW-1** killed bacteria by disrupting cell membranes, confirmed with TEM, OM permeabilization, and membrane depolarization though without apparent drug-developed resistance. Meanwhile, it displayed good selectivity with low hemolytic toxicity. Taken together, **YW-1** showed good therapeutic potential, and it could be a candidate to solve the problem of AMR.

## Data Availability

All the data presented in this study are available on request from the corresponding authors.

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
