# Peer review of "Short, Lipidated Dendrimeric γ-AApeptides as New Antimicrobial Peptidomimetics"

_ijms, 2023, doi:10.3390/ijms24076407_

Round 1

Reviewer 1 Report

This manuscript describes a new antimicrobial lipidated γ-AApeptides. The authors have developed dendrimeric γ-AApeptides which have antibacterial activity for gram-positive and gram-negative bacteria. The design of short lipidated γ-AApeptides in this stud is unique, in this point, the study has originality and contributes to the field. However, the manuscript includes several issues that should be improved to be published.

Major points

1) Table 1: When the author performed the assay several times, the results should be described in the mean and SEM values.

2) Figure 2: In the left graph, the plots of 2x and 4x are overlapped. Is the experiment reliable?

3) Figure 3B: Why did the OD values indicate a sudden increase between 2 x MIC and 3x MIC?

4) Figure 4: The density of bacteria seems extremely lower in the presence of YW-1 than in the absence of YW-1. Did the authors use the same condition about the number of bacteria?

Minor points

5) The preparation of the short lipidated dendrimeric γ-AApeptides should be described.

6) Figure 6: In the PDF file, a part of figure 6B is cut off

Author Response

Thanks for your comments. Please find attached our response.

Reviewer 2 Report

This manuscript details the development of lipidated dipeptides as new antimicrobials. These structures are based on host-defense peptides, and therefore are also decorated with a number of amines which provide charged functional groups to take the place of polar amino acid sidechains in the natural systems. These peptides show broad-spectrum activity against both Gram-positive bacteria and Gram-negative bacteria, including MRSA. Mechanistic studies show that these peptides are likely exerting their cell killing by the permeabilization of bacterial membranes. Interestingly, the authors claim that resistance to these antibiotics does not develop readily. This report would seem to interest those working in the infectious disease area.

No details of the synthesis or characterization of eleven lipidated dipeptides (YW-1 through YW-11) is provided. Without this information it is impossible to determine if the structures of these molecules are correct and/or if the tested compounds were at all pure. Proof of identity and purity should be provided on the final products at the very least. A synthetic scheme detailing how the molecules were accessed and what intermediates were involved would also be appropriate, with details of identity and purity of the intermediates.

With the compound characterization data missing, I can only suggest a major revision as it needs to be provided before publication. In addition, there are a couple of minor formatting issues listed below, these should be addressed as well.

In Figure 2, panels A and B should be labelled for clarity.

The second panel of Figure 6 has been cut off on my copy of the manuscript, so it's difficult to determine if the findings that are claimed in the text are supported by the data.

Author Response

(The authors gave the same response as above.)

Reviewer 3 Report

The manuscript by Wang et al described the design of lipidated dendrimeric γ-AApeptides as a new class of antimicrobial agents. The authors identified lead compound, YW-1, which displayed potency towards both Gram-positive and Gram-negative bacteria. Further, the mechanistic studies revealed that the identified compound could kill bacteria rapidly through the permeabilization of bacterial membranes confirmed with TEM, OM permeabilization, and membrane depolarization.

The manuscript has been written well. However there are few issues which needs to be addressed before the manuscript can be published:

1. The author should describe in detail the rational behind the design of new peptide compounds as very limited discussion is provided.

2. No information is provided on the synthesis of peptides in the methods sections.

3. The authors should provide HPLC, NMR and mass spectrometry data in the supporting information.

4. The quality of figures need improvement; for example a part of figure 6b is cut. 

Author Response

(The authors gave the same response as above.)

Round 2

Reviewer 1 Report

For the current version, I recommend it for publication on IJMS.

Author Response

We thank the reviewer for their comments

Reviewer 2 Report

The authors have responded to my concerns, and the active compounds now have a synthetic scheme and data to support their structure and purity. The errors with regard to the figures have also been addressed.

Author Response

We thank reviewer for their comments

Reviewer 3 Report

The manuscript has improved from the previous version but still there are flaws with the data reported in the supporting information.

1.     There is no reference “20” present as mentioned in the general synthesis of compound in the supporting information.

2.     The 13C spectrum of YW1 is of poor quality. There are four carbonyl (-C=O) peaks expected. But authors have mentioned five carbonyl peaks in the reported data. Is there any sort of isomerism happening? Further, there are two peaks at 116.39 and 118.33 which corresponds to aromatic carbons but there is no aromatic ring present in compound YW1.

3.      Given the high quality of HPLC chromatograph the NMR data is not reflecting that especially carbon data.

4.     The author should provide the traces of mass data instead of a table.

Author Response

We thank reviewer for their comments. Please find attached our response.

Round 3

Reviewer 3 Report

I am satisfied with the authors response.